# Generalization with Lossy Affordances: Leveraging Broad Offline Data for Learning Visuomotor Tasks

**Kuan Fang, Patrick Yin, Ashvin Nair, Homer Walke, Gengchen Yan, Sergey Levine**

University of California, Berkeley

**Abstract:** The use of broad datasets has proven to be crucial for generalization for a wide range of fields. However, how to effectively make use of diverse multi-task data for novel downstream tasks still remains a grand challenge in reinforcement learning and robotics. To tackle this challenge, we introduce a framework that acquires goal-conditioned policies for unseen temporally extended tasks via offline reinforcement learning on broad data, in combination with online fine-tuning guided by subgoals in a learned lossy representation space. When faced with a novel task goal, our framework uses an affordance model to plan a sequence of lossy representations as subgoals that decomposes the original task into easier problems. Learned from the broad prior data, the lossy representation emphasizes task-relevant information about states and goals while abstracting away redundant contexts that hinder generalization. It thus enables subgoal planning for unseen tasks, provides a compact input to the policy, and facilitates reward shaping during fine-tuning. We show that our framework can be pre-trained on large-scale datasets of robot experience from prior work and efficiently fine-tuned for novel tasks, entirely from visual inputs without any manual reward engineering. [1]

**Keywords:** Reinforcement Learning, Representation Learning, Planning

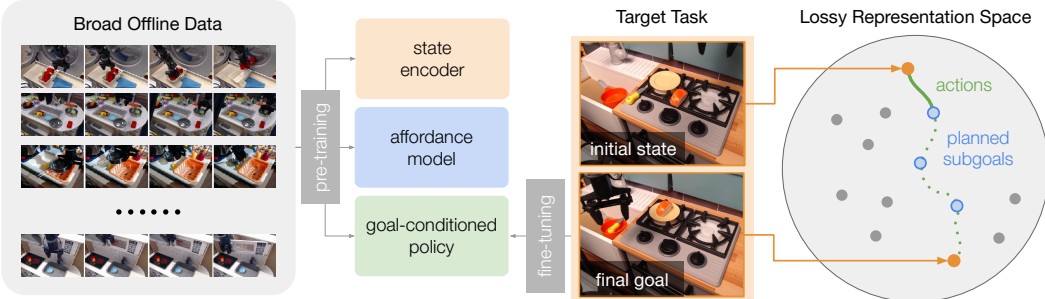

Figure 1: **Fine-Tuning with Lossy Affordance Planner (FLAP).** Our framework leverages broad offline data for new temporally extended tasks using learned lossy representations of states and goals. We pre-train a state encoder, and affordance model, and a goal-conditioned policy on the offline data collected from diverse environments and fine-tune the policy to solve the target tasks without any explicit reward signals. To provide guidance for the policy, subgoals are planned in the lossy representation space given visual inputs.

## 1 Introduction

Learning-based methods can enable robotic systems to automatically acquire large repertoires of behaviors that can potentially generalize to diverse real-world scenarios. However, attaining such generalizability requires the ability to leverage large-scale datasets, which presents a conundrum when building general-purpose robotic systems: How can we tractably endow robots with the desired skills if each behavior requires a laborious data collection and lengthy learning process to master? In much the same way that humans use their past experience and expertise to acquire new

---

[1]Project webpage: sites.google.com/view/project-flap

6th Conference on Robot Learning (CoRL 2022), Auckland, New Zealand.

skills more quickly, the answer for robots may be to effectively leverage prior data. But in the setting of robotic learning, this raises a number of complex questions: What sort of knowledge should we derive from this prior data? How do we break up diverse and uncurated prior datasets into distinct skills? And how do we then use and adapt these skills for solving novel tasks?

To learn useful behaviors from data of a wide range of tasks, we can employ goal-conditioned reinforcement learning (GCRL) [1, 2], where a policy is trained to attain a specified goal (e.g., indicated as an image) from the current state. This makes it practical to learn from previously collected large datasets without explicit reward annotations and to share knowledge across tasks that exhibit similar physical phenomena. However, it is usually difficult to learn goal-conditioned policies that can solve temporally extended tasks in zero shot, as such policies are typically only effective for short-horizon goals [3]. To solve the target tasks, we would need to transfer the learned knowledge to the testing environment and efficiently fine-tune the policy in an online manner.

Our proposed solution combines representation learning, planning, and online fine-tuning. The key intuition is that, if we can learn a suitable representation of states and goals that generalizes effectively across environments, then we can plan subgoals in this representation space to solve long-horizon tasks, and also leverage it to help finetune the goal-conditioned policies on the new task online. Core to this planning procedure is the use of *affordance models* [4, 5], which predict potentially reachable states that can serve as subgoal proposals for the planning process. Good state representations are necessary for this process: (1) as inputs and outputs for the affordance model, which must generalize effectively across tasks and domains (since if the affordance model doesn't generalize, it can't provide guidance for policy fine-tuning); (2) as inputs into the policy, so as to facilitate rapid adaptation; (3) as measures of state proximity for use as *reward functions* for fine-tuning the policy, since well-shaped rewards are essential for rapid online training but notoriously difficult to obtain without manual engineering.

To this end, we propose Fine-Tuning with Lossy Affordance Planner (FLAP), a framework that leverages diverse offline data for learning representations, goal-conditioned policies, and affordance models that enable rapid fine-tuning to new tasks in target scenes. As shown in Fig. 1, our lossy representations are acquired during the offline pre-training process for the goal-conditioned policy using a variational information bottleneck (VIB) [6]. Intuitively, this representation learns to capture the minimal information necessary to determine if and how a particular goal can be reached from a particular state, making it ideally suited for planning over subgoals and providing reward shaping during fine-tuning. When a new task is specified to the robot with a goal image, we use an affordance model that predicts reachable states in this learned lossy representation space to plan subgoals for prospectively accomplishing this goal. During the fine-tuning process, the goal-conditioned policy is fine-tuned on each subgoal given the informative reward signal computed from the learned representations. Both the offline and online stage in this process operates entirely from images and does not require any hand-designed reward functions beyond those extracted automatically from the learned representation. Building the components based on the prior work [5, 7], we demonstrate that the particular combination of lossy representation learning, goal-conditioned policies, and planning-driven fine-tuning can enable performance that significantly exceeds that of prior methods. We evaluate our method in both real-world and simulated environments using previously collected offline data [8, 7] for learning novel robotic manipulation tasks. Compared to baselines, the proposed method achieves higher success rates with fewer online fine-tuning iterations.

## 2 Preliminaries

**Goal-conditioned reinforcement learning.** We consider a goal-conditioned reinforcement learning (GCRL) setting with states and goals as images. The goal-reaching tasks can be represented by a Markov Decision Process (MDP) denoted by a tuple $M = (\mathcal{S}, \mathcal{A}, \rho, P, \mathcal{G}, \gamma)$ with state space $\mathcal{S}$, action space $\mathcal{A}$, initial state probability $\rho$, transition probability $P$, a goal space $\mathcal{G}$, and discount factor $\gamma$. Each goal-reaching task can be defined by a pair of initial state $s_0 \in \mathcal{S}$ and desired goal $s_g \in \mathcal{G}$. We assume states and goals are defined in the same space, i.e. $\mathcal{G} = \mathcal{S}$. By selecting the action $a_t \in \mathcal{A}$ at each time step $t$, the goal-conditioned policy $\pi(a_t|s_t, s_g)$ aims to reach a state $s$ such that $d(s - s_g) \leq \epsilon$, where $d$ is a distance metric and $\epsilon$ is the threshold for reaching the goal. We use the sparse reward function $r_t(s_{t+1}, s_g)$ which outputs 0 when the goal is reached and $-1$ otherwise and the objective is defined as the average expected discounted return $\mathbb{E}[\Sigma_t \gamma^t r_t]$.

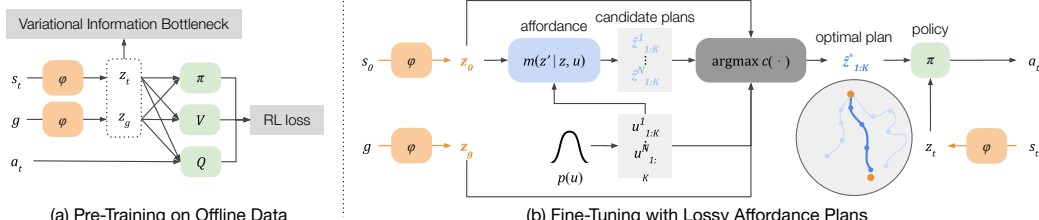

(a) Pre-Training on Offline Data  (b) Fine-Tuning with Lossy Affordance Plans

Figure 2: **FLAP Architecture.** (a) We pre-train the representation with RL on the offline data. We first encode the initial state and the goal and then compute the RL loss with policy $\pi$, the value function $V$, and the Q-function $Q$. (b) We use planned subgoals to guide the policy during online fine-tuning (right). Given the encodings of the initial state and the final goal, subgoal sequences are recursively generated by the affordance model in the learned representation space with latent codes sampled from the prior $p(u)$. The optimal subgoal sequence $\hat{z}^*_{1:K}$ is then selected according to Eq. 6 for guiding the goal-conditioned policy.

**Planning with affordance models.** When learning to solve long-horizon tasks, subgoals can significantly improve performance by breaking down the original problem into easier short-horizon tasks. We can use sampling-based planning to find a suitable sequence of $K$ subgoals $\hat{s}_{1:K}$, which samples multiple candidate sequences and chooses the optimal plan based on a cost function. In high-dimensional state spaces, such a planning process can be computationally intractable, since most sampled candidates will be unlikely to form a reasonable plan. To facilitate planning, we would need to focus on sampling subgoal candidates that are realistic and reachable from the current state. Following the prior work [5, 7], we use affordance models to capture the distribution of reachable future states and recursively propose subgoals conditioned on the initial state of each task. The affordance model can be defined as a generative model $m(s'|s, u)$, where $u$ is the latent representation that captures the information of the transition from the current state $s$ to the goal $s'$. It can be trained in a conditional variational autoencoder (CVAE) [9] paradigm using the encoder $q(u|s, s')$ to estimate the evidence lower bound (ELBO) [10].

**Offline pre-training and online fine-tuning.** Offline reinforcement learning pre-trains the policy and value function on a dataset $\mathcal{D}_{\text{offline}}$, consisting of previously collected experiences $(s^i_t, a^i_t, r^i_t, s^i_{t+1})$, where $i$ and $t$ are the index of the trajectory and time step. In this work, we assume $\mathcal{D}_{\text{offline}}$ does not include data of the target task. To solve the target task, the pre-trained policy is then fine-tuned by exploring the environment and collecting online data $\mathcal{D}_{\text{online}}$. Following Fang et al. [7], we first pre-train both the policy $\pi$ and the affordance model $m$ on $\mathcal{D}_{\text{offline}}$, and then use the subgoals planned using the affordance model to guide the online exploration of $\pi$. In the online phase, we freeze the weight of $m$ and fine-tune $\pi$ on a combination of $\mathcal{D}_{\text{offline}}$ and $\mathcal{D}_{\text{online}}$. While our contribution is orthogonal to the specific choice of the offline RL algorithm, we use implicit Q-learning [11] which is well-suited to online fine-tuning using an expectile loss to construct value estimates and advantage weighted regression to extract a policy.

## 3 Pre-Training and Fine-Tuning with a Lossy Affordance Planner

The objective of our method is to enable a robot to leverage diverse offline data to efficiently learn how to achieve new goals, potentially in new scenes. While we do not require the offline dataset to contain data of the target tasks, two assumptions are made for enabling effective possible generalization. First, we will consider tasks that require the robot to compose previously seen behaviors (e.g., pushing, grasping, and opening) in order to perform a temporally extended task that sequences these behaviors in a new way (e.g., pushing an obstruction out of the way and then opening a drawer). Second, we will consider tasks that require the robot to perform behaviors that resemble those in the prior data, but with new objects that it had not seen before, which presents a generalization challenge for the policies and the affordance models.

Our method is based on offline goal-conditioned reinforcement learning and subgoal planning with affordance models. To solve temporally extended tasks with novel objects, we construct the goal-conditioned policy and the affordance model in a learned representation space of states and goals that picks up on task-relevant information (e.g., object identity and location), while abstracting away unnecessary visual distractors (e.g., camera pose, illumination, background). In this section, we first describe a paradigm for jointly pre-training the goal-conditioned policy and the lossy representation through offline reinforcement learning. Then, we describe how to construct an affordance model in this lossy representation space for guiding the goal-conditioned policy with planned subgoals.

Finally, we describe how we utilize the lossy affordance model in a complete system for vision-based robotic control to set goals and finetune policies online.

## 3.1 Goal-Conditioned Reinforcement Learning with Lossy Representations

To capture the task-relevant information of states and goals, we learn a parametric state encoder $\phi(z|s)$ to project both states and goals from the high-dimensional space (e.g., RGB images) $\mathcal{S}$ into a learned representation space $\mathcal{Z}$. We use $z_t$ and $z_g$ to denote the representation of the state $s_t$ and the goal $g$, respectively. In the learned representation space, we construct the goal-conditioned policy $\pi(a|z_t, z_g)$, as well as the value function $V(z_t, z_g)$ and Q-function $Q(z_t, z_g, a)$. Both $V$ and $Q$ are used for training the policy via the offline RL algorithm, which in our prototype is IQL [11], and $V$ is also used for selecting subgoals in the planner, which will be discussed in Sec. 3.2. When training $\phi$, $\pi$, $V$, and $Q$ to optimize the goal-reaching objective described in Sec. 2, $\phi$ would need to extract sufficient information for selecting the action to transit from $s_t$ to $g$ and estimating the required discounted number of steps.

We jointly pre-train the representation with reinforcement learning on the offline data as shown in Fig. 2. The original IQL objective optimizes $\mathcal{L}_{\text{RL}} = \mathcal{L}_\pi + \mathcal{L}_Q + \mathcal{L}_V$ as described in Kostrikov et al. [11]. To facilitate generalization of the pre-trained policy and the affordance model, we would like to abstract away redundant domain-specific information from the learned representation. For this purpose, we add a variational information bottleneck (VIB) [6] to the reinforcement learning objective, which constrains the mutual information $I(s_t; z_t)$ and $I(g; z_g)$ between the state space and the representation space by a constant $C$. The joint training of $\phi$, $\pi$, $V$ and $Q$ can be written as an optimization problem using $\theta$ to denote the model parameters:

$$\max_\theta \ \mathcal{L}_{\text{RL}} \qquad \text{s.t.} \ \ I(g; z_g) \leq C, \quad I(s_t; z_t) \leq C, \text{for } t = 0, ..., T \tag{1}$$

Following Alemi et al. [6], we convert this into an unconstrained optimization problem by applying the bottleneck on the Q-function representation and directly selecting the Lagrange multiplier $\alpha$, resulting in the full VIB objective:

$$\mathcal{L} \ = \ \mathcal{L}_{\text{RL}} - \alpha D_{KL}(\phi(z_t|s_t) \parallel p(z)) - \alpha D_{KL}(\phi(z_g|g) \parallel p(z)) \tag{2}$$

where we use the normal distribution as $p(z)$, the prior distribution of $z$.

By optimizing this objective, we obtain lossy representations that disentangle relevant factors such as object poses from irrelevant factors such as scene background. We exploit this property of the learned representations in several ways: learning policies and affordance models that generalize from prior data, having a prior for optimization, and as a distance metric for rewards during finetuning. Next, we will describe how we generate subgoals utilizing the bottleneck representation.

## 3.2 Composing Subgoals using Lossy Affordances

The effectiveness of using subgoals to guide the goal-conditioned policy relies on two conditions. First, each pair of adjacent subgoals needs to be in the distribution of $\mathcal{D}_{\text{offline}}$ so that the pre-trained goal-conditioned policy will have a sufficient success rate for the transition. Second, the subgoals should break down the original task into subtasks of reasonable difficulties. As shown in Fig. 2, we devise an affordance model and a cost function to plan subgoals that satisfy these conditions and use the learned lossy representation to facilitate generalization to novel target tasks.

Instead of sampling goals from the original high-dimensional state space, we propose subgoals $\hat{z}_{1:K}$ in the learned lossy representation space. For this purpose, we learn a lossy affordance model to capture the distribution $p(z'|z)$, where $z$ is the representation of a state $s$ and $z'$ corresponds to the future state $s'$ that is reachable within $\Delta t$ steps. The lossy affordance model can be constructed as a parametric model $m(z'|z, u)$, where $u$ is a latent code that represents the transition from $z$ to $z'$. Given a sequence of sampled latent codes $u_{1:K}$, we can recursively sample the $k$-th subgoal $\hat{z}_k$ in the sequence $\hat{z}_{1:K}$ with $m$ by taking $\hat{z}_{k-1}$ and $u_k$ as inputs, where we denote $\hat{z}_0 = z_0$.

The lossy affordance model is pre-trained on the offline dataset $\mathcal{D}_{\text{offline}}$. Given each pair $(s, s')$ sampled from $\mathcal{D}_{\text{offline}}$, the pre-trained encoder $\phi$ produces a distribution of lossy representation $\phi(z'|s')$. Given the sampled representation $z$, we would like the affordance model to propose representations that follow the distribution $\phi(z'|s')$. Using $p_m(z'|z)$ to denote the marginalized distribution of the

lossy affordance model, the learning objective can be defined as the KL-divergence:

$$D_{KL}(p_m(z'|z) \| \phi(z'|s')) \tag{3}$$

Under the conditional variational autoencoder (CVAE) [9] paradigm, we define the CVAE encoder $q(u|z, z')$ to infer the latent code $u$. The ELBO [10] of this objective can be written as:

$$\mathbb{E}_{z \sim \phi, u \sim q} [D_{KL}(m(z'|z, u) \| \phi(z'|s'))] - \beta \mathbb{E}_{z, z' \sim \phi} [D_{KL}(q(u|z, z') \| p(u))] \tag{4}$$

Given the trained affordance model, the sampling-based planning can be conducted in the lossy representation space. Given the initial state $s_0$ and the final goal $g$, we first project them into latent representations $z_0$ and $z_g$ using $\phi$. Then, we would like to find the optimal sequence of subgoals in the representation space that (1) reaches the final goal $z_g$, (2) consists of a sequence of subgoals that the RL policy believes are easy to reach, and (3) where each subsequent subgoal has a high probability under the affordance model given the previous one, indicating that the sequence of subgoals is physically plausible. We use the Euclidean distance in the lossy representation space to measure if the goal is reached. We use the value function to estimate the difficulty of the transition from one subgoal to another and require the estimated value of the transition between adjacent subgoals fall into a specified range $[V_{\min}, V_{\max}]$. The probability of the transition can be measured by the prior probability $p(u)$ of the latent code $u$ used for generating the subgoal from the affordance model, with low-probability transition latent $u$ corresponding to implausible transitions. To summarize, the constrained optimization problem that our planning algorithm aims to solve is given by

$$\begin{aligned} \text{minimize} \quad & \|z_g - \hat{z}_K\| \\ \text{subject to} \quad & V_{\min} \le V(\hat{z}_{k-1}, \hat{z}_k) \le V_{\max}, p_{\min} \le p(u_k), \text{for } k = 1, ..., K \end{aligned} \tag{5}$$

where we denote $V_k = V(\hat{z}_{k-1}, \hat{z}_k)$. The cost function can thus be written as

$$c(z_0, z_g, u_{1:K}) = \|z_g - \hat{z}_K\| + \sum_{k=1}^{K} \left( \eta_1 (V_{\min} - V_k)^+ + \eta_2 (V_k - V_{\max})^+ + \eta_3 \log p(u_k) \right) \tag{6}$$

where $\eta_1$, $\eta_2$, and $\eta_3$ are the Lagrange multipliers and we use $(\cdot)^+$ to denote $\max(\cdot, 0)$ which truncates the negative values to zero. After sampling candidate subgoal sequences in the lossy representation space, we evaluate each candidate using the cost function. Model predictive path integral (MPPI) [12] is used to iteratively optimize the plan through importance sampling.

### 3.3 Fine-Tuning with Planned Subgoals

Finally, when the task is different from the training distribution, the learned offline policy may not generalize perfectly, so we fine-tune with online interaction on a target task specified by a goal image $g$. The affordance model is used to generate subgoals during this phase. Given the encoding of the goal image $z_g$, we optimize subgoals $\hat{z}_{1:K}$ according to Eq. 6. During a trajectory, the subgoals are fed into the policy sequentially, executing action $a_t \sim \pi(a|s_t, \hat{z}_k)$ switching to a new subgoal representation $z_k$ from the plan every $h$ steps. During online fine-tuning, we freeze the weights of the state encoder and only update the policy, the value function, and the Q-function.

For online fine-tuning, we need to evaluate rewards. While GCRL algorithms can operate from only relabeled sparse rewards, shaped reward functions can assist in improving performance. But simple rewards such as the Euclidean distance between $s$ and $g$ as in Sec. 2 can often be misleading for determining the goal-reaching success condition, since a small change in the environment can often result in huge difference in the corresponding image observation. On the other hand, the learned lossy representation is trained to make distances in the latent space meaningful, and can thus be more robust to such changes. We leverage this property during fine-tuning: in addition to serving as the input to the policy and value functions, the learned representation is used to define a more informative reward function. Therefore, once the encoder $\phi$ is pre-trained and fixed, we re-define the reward function to be the thresholded cosine similarity $r(s, g) = \mathbb{1}\{(z_g \cdot z_t)/(\|z_g\|\|z_t\|) < \epsilon\}$ instead, which provides a more informative reward signal during fine-tuning.

## 4 Experiments

In our experiments, we aim to answer the following questions: (1) Can the proposed method produce meaningful subgoals in the learned latent space? (2) Can the subgoals generated by the lossy affordances facilitate learning new tasks in new scenes? (3) How much does offline data from other environments and tasks actually aid in learning the new task using FLAP?

## 4.1 Experimental Setup

**Environments and tasks.** Our experiments are conducted in table-top manipulation environments using low-level visuomotor control, with observations and goals corresponding to RGB images captured by a third-person camera. We use a real-world WidowX robot operating in kitchen-themed environments at a frequency of 5 Hz through 7-DoF control. The action space is defined as the translation, rotation, and finger position of the gripper, without using any hard-coded motion primitive. In each target task, the robot is asked to reach the goal in 200 steps by interacting with multiple objects unseen in the environment. The target tasks include moving a metal pot from one stove to another (Task A), moving a colander onto the stove and picking and placing a stuffed toy into the pot (Task B), and placing a sushi on a plate and then dropping a knife into a pan in the sink

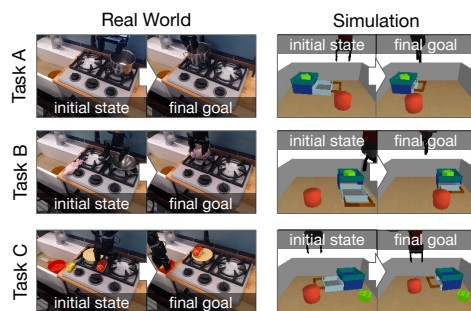

Figure 3: **Target tasks.** Three multi-stage tasks are evaluated in the real world and the simulation. In each target task, the robot needs to strategically interact with the environment. The initial state and the desired goal state are shown for each task.

(Task C). We also use a simulated environment for additional ablations and experiments based on prior work [13]. The simulated target tasks require sequential interactions with the environment such as opening/closing the drawer, sliding the cylinder, and picking and placing the block. To complete these tasks, the robot needs to reason about the objects' affordances and spatial relations from the visual inputs and strategically interact with the environment in a feasible sequential order.

**Offline Datasets.** In the real world, we use the Bridge Dataset [8] which consists of 11,980 demonstration trajectories in 10 different environments with a variety of objects. Note that this dataset was not collected or designed specifically for our method, but is sourced directly from prior work (where it was used for imitation learning). In simulation, the offline data consists of 12,000 trials in 6 scenes, which differ in terms of object texture and camera pose.

**Prior methods and comparisons.** We compare FLAP to four other approaches. **Model-Free** is broadly representative of prior work that learns a goal-conditioned policy by directly feeding the final goal without using any subgoals [5, 13, 14]. **LEAP** [15], **GCP** [16], and **PTP** [7] learn to plan subgoals in the original high-dimensional state space, instead of a learned lossy representation space as in our method. We also compare our full model **Ours (Broad Data)**, which is pre-trained on the previously discussed diverse prior datasets, with an ablation **Ours (Target Data)**, which is trained only on a subset of that data from our target domain. This baseline allows us to evaluate the importance of pre-training on a large and diverse dataset. All methods use the same neural network architectures, and all methods except for the ablation are pre-trained on the same prior data.

## 4.2 Comparative Results

**Real-world evaluation.** The results for a real-world evaluation with fine-tuning on three new tasks are presented in Table 1. The results show that across all three tasks, our full system consistently leads to significant improvement from fine-tuning, and indeed is the only method to produce improvement across all tasks. Note that these tasks involve multiple distinct stages, presenting a major challenge to standard goal-conditioned RL methods, as indicated by the poor performance of the **Model-Free** baseline both before and after fine-tuning. The **PTP** [7] baseline generally struggles in these domains due to their visual complexity, and from a cursory examination, the generated image-space subgoals are of low quality and blurry, which likely leads to poor subgoal guidance for the goal-conditioned policy. Without prior data pre-training , both the policy and the affordance model generalize poorly and generally fail to make progress, which indicates that much of the benefit of our approach comes from being able to effectively leverage the diverse prior data pre-training.

**Simulation.** We report the average success rate during the online fine-tuning phase in Fig. 4. In all the three tasks, our full system consistently outperforms the prior methods and baselines. At the beginning of fine-tuning, the success rate is close to zero for all the methods. Using the subgoals planned with the lossy affordance models, FLAP efficiently improves the success rate to 65.6%, 62.9%, and 89.5%. Without the planned subgoals (**Model-Free**), the robot often ends up performing the individual steps of the task in the wrong order, or is forced to rely on random action exploration

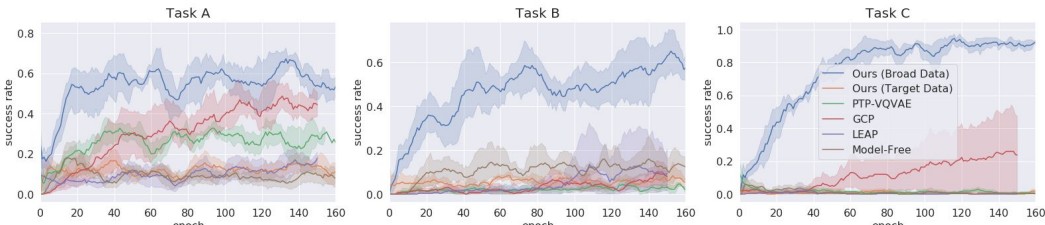

Figure 4: **Quantitative comparison in simulation.** The success rate averaged across 5 runs during fine-tuning is shown with the shaded region indicating the standard deviation. Each epoch consists of 5 trajectories.

Table 1: **The real-world success rates after offline pre-training and then after online fine-tuning.**

| Task | Model-Free | PTP | Ours (Target Data) | Ours (Broad Data) |
|------|------------|-----|--------------------|--------------------|
| | Offline → Online | Offline → Online | Offline → Online | Offline → Online |
| Task A | 0.0% → 0.0% | 0.0% → 25.0% | 0.0% → 12.5% | 12.5% → **75.0%** |
| Task B | 0.0% → 0.0% | 0.0% → 0.0% | 25.0% → 25.0% | 25.0% → **62.5%** |
| Task C | 0.0% → 0.0% | 0.0% → 0.0% | 12.5% → 12.5% | 25.0% → **50.0%** |

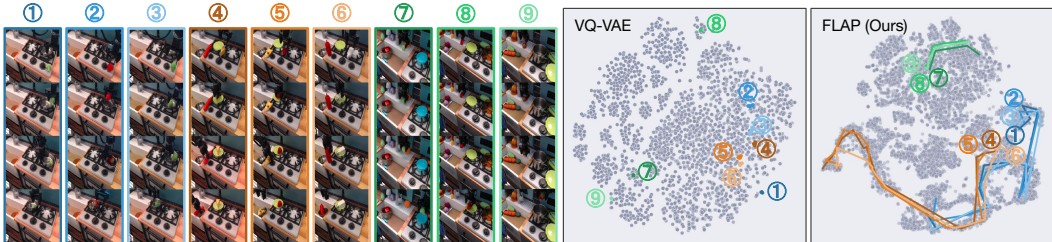

Figure 5: **Visualization of Learned Lossy Representations.** We compare our learned lossy representations with VQ-VAE using t-SNE. We sample 9 diverse trajectories from 3 different tasks (indicated by different colors) and plot the encoded trajectories. The trajectories and their initial states are indicted by the corresponding colors and number. FLAP demonstrates to have a more informative structure, as discussed in Sec. 4.3.

and fails to make progress. Especially in Task C, the baselines always attempt to open the drawer before moving the obstacle out of the way, which leads to failure. Prior methods such as PTP that generate goals in the high-dimensional image space, rarely find suitable subgoals for the novel tasks. When trained on only the target data (i.e., **Ours (Target Data)**), both the policy and the affordance model generalize poorly and generally fail to make progress, which indicates that much of the benefit of our approach comes from being able to effectively leverage the diverse prior data pre-training.

## 4.3   Learned Lossy Representations

We visualize the learned lossy representations in the real world using t-SNE [17] in Fig. 5, comparing them to a representation learned via a VQ-VAE model [18], as used by PTP [7]. Image observations in the target scene are sampled from the offline dataset and encoded using our state encoder and VQ-VAE, respectively. Each encoded representation is plotted as a grey dot. From 3 different tasks, we sample 9 trajectories with varied camera poses, lighting conditions, objects, and motions. We subsample the trajectories every 5 time steps, then plot the encoded trajectories overlaid on the t-SNE plots, with the number showing the start state of each one. Different colors indicate different trajectories and trajectories of the same semantic meaning are marked by similar colors.

The embedding illustrates several interesting differences between our learned lossy embedding and the task-agnostic VQ-VAE embedding. Although both embeddings exhibit some grouping or clumps (corresponding to visually and semantically distinct scenes), our representations appear to dedicate more representational power to functionally relevant degrees of freedom as demonstrated in in Fig. 5. Specifically, when using the VQ-VAE encodings, each trajectory collapses into a different clique regardless whether they are from the same type or how much the scene changes in each trajectory. However, when in our learned lossy representation space, each trajectory traverses a larger region. The second half of trajectories 1-3 and the first half of trajectories 4-6 are both moving objects from the right stove to the left stove, and interestingly we see that they intersect in the t-SNE plot.

# 5 Related Work

There is a growing body of work on learning policies from broad and diverse datasets in control and robotics using offline RL [19, 20, 21, 22, 23, 24, 25, 26, 27, 28]. Learned offline policies can also be fine-tuned with additional online interaction [29, 30, 31, 5, 32, 33]. While most of these methods focus on learning from offline data of the same task, this paper aims to utilize data of diverse environments and tasks to facilitate fine-tuning for novel tasks. A wide range of methods have been proposed for multi-task learning [34, 35] and meta learning [36, 37]. In contrast to these methods, we do not assume the task labels or explicit reward functions are provided in the offline data, instead adopting a goal-conditioned approach that can utilize diverse datasets without any explicit task labels or rewards. Goal-conditioned RL [1, 2] aims to learn policies that can flexibly solve multiple tasks by taking different goals as inputs. Through goal relabeling [38], a goal-conditioned policy can be trained with self-supervision in absence of meticulously designed reward functions [39, 40, 41, 42, 43, 44, 45, 46, 47, 48]. To guide the exploration of goal-conditioned policy, several methods use goals produced by a goal generator or affordance model that learns to capture the distribution of reachable goals [14, 49, 13, 5, 50]. Compared to this prior work, we show that we can utilize broad datasets by utilizing representation learning within goal-conditioned RL. Moreover, by utilizing planning, we can significantly extend the capabilities of goal-conditioned RL. Recent works have also proposed to plan sequences of subgoals for learning long-horizon tasks [15, 3, 16, 7]. These methods successfully improve the performance of goal-conditioned RL, but have significant shortcomings in utilizing broad datasets to learn novel tasks. Nasiriany et al. [15] plans in the latent space of a variational autoencoder [10] for goal-conditioned RL in a single environment, but does not handle diverse prior data. Nonparametric methods [3] utilize graph-based planning, which makes them difficult to adapt for learning unseen tasks from data where states visited for the new task have not been traversed. Instead of generating and planning subgoals in the original state space, our method learns a representation of goals to facilitate generalization across domains. A number of methods have been proposed for learning latent representations of states for control and robotics [51, 52, 53, 54, 55, 56, 57, 58, 59]. While most focus on learning compact representations for learning the policy or the dynamics model, our method learns representations in an offline goal-conditioned reinforcement learning setting and uses the learned representations to facilitate subgoal planning across domains. Shah et al. [60], Shah and Levine [61] use a mutual information objective learns representation of the transition for selecting the goals to guide exploration. Instead of proposing subgoals from previously visited states or a prior distribution, our method learns a generative model of the latent goals to plan subgoals for target tasks do not exist in the offline data.

# 6 Conclusion

We presented FLAP, a framework that leverages broad offline data to more effectively learn new temporally extended tasks. Our framework uses an affordance model to provide planned subgoals to guide a goal-conditioned policy, which is then finetuned for the new task. Our method trains a lossy representation of states and goals from the offline data, which then provides compact inputs to the policy, facilitates subgoal planning, and shapes reward signals during the online fine-tuning process. Our framework can effectively utilize offline data collected from diverse environment and tasks, and because of the use of GCRL, it can be pre-trained on this offline data without any hand-specified reward signal. Our results show that FLAP significantly improves over alternative designs for such a system, and attains good results even in visually complex real-world settings.

**Limitations.** The main limitation of our method is that the offline data and the target task need to share enough similarity for effective knowledge transfer. First, the state spaces need to have the same dimension, otherwise the state encoder pre-trained on the offline dataset cannot be applied for the target task. Second, the offline data needs to contain trajectories with structurally similar behaviors to those that are required to complete the new task, so that the affordance model can sample meaningful subgoals after being pre-trained on the offline data.

**Acknowledgement.** We acknowledge the support by the Office of Naval Research, ARL DCIST CRA W911NF-17-2-0181, ARO W911NF-21-1-0097, and AFOSR FA9550-22-1-0273. We would like to thank Frederik Ebert, Dibya Ghosh, and Dhruv Shah for their constructive feedbacks. And we would like to thank Abraham Lee for helping with the data collection.

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
