# OpenReview forum: "Generalization with Lossy Affordances: Leveraging Broad Offline Data for Learning Visuomotor Tasks"
_robot-learning.org/CoRL/2022/Conference — CoRL 2022 Oral_

### Official Review · Reviewer_xMTo · 2022-07-13

**Originality:** Very Good
**Technical Quality:** Excellent
**Clarity Of Presentation:** Excellent
**Impact:** 4

**Recommendation:**

Strong Accept: I recommend accepting the paper and will argue for my recommendation even if other reviewers hold a different opinion.

**Summary:**

This paper deploys a reinforcement learning (GCRL) framework, where a policy is trained to attain a specified goal (indicated as an image) from the current state. This is a very practical setting, where the agent can learn from offline dataset of images without explicit reward annotations. The proposed framework, Fine-Tuning with Lossy Affordance Planner (FLAP), leverages diverse offline data for learning representations, goal-conditioned policies, and core to the planning procedure, the affordances. Affordance models predict potentially reachable states that can serve as subgoal proposals for the planning process. This is useful when planning long-horizon tasks, subgoals can assist the learning by breaking down the original problem into easier short-horizon tasks. This also enables sharing the same dataset for learning various tasks that are sub-skills of the original task.



**Issues:**

I have the following questions/comments for the authors:

- Missing reference for implicit Q-learning  (IQL) for training offline RL algorithm. Perhaps you can provide a brief overview. I had to look it up.
- Figure 5: Visualization of Learned Lossy Representations. How does one know which region corresponds to what task and how were the sub-task trajectories followed? I wasn't sure if these trajectories were determined by hand or automatically somehow.
- Perhaps some discussion about the baselines and how this method is expanding on them, or what is missing from existing methods would be useful to a reader unfamiliar with these methods. They are only referenced and the final results are shown (maybe you removed this due to lack of space, but for sure to be included in the full paper version).
- Why is there such a vast difference between your method Ours(Broad data) and (Target Data) ? I understand this was trained on a a subset of that data from our target domain, but the Target Data almost does not learn at all. Which for me raises the question of generalizability of the (Broad data) version. I understand you run real robot experiments on real RGB images, but as far as I understood, there is no zero-shot generalization here, but rather fine-tuning on the real data?
- For the limitation, the authors mention "the state spaces need to have the same dimension, otherwise the state encoder pre-trained on the offline dataset cannot be applied for the target task."
I dont see this as a huge problem, am I wrong? Since you are using images as inputs. Does the input also include proprioceptive states of the robot/task too?
- I like this observation/motivation made by the authors "By optimizing this objective, we obtain lossy representations that disentangle relevant factors such as object poses from irrelevant factors such as scene background." But is there a way to support this claim? Just by the results presented here, its not immediately clear to me that this is the case.


**Quality Of The Limitations Section:**

Limitations are addressed clearly

**Reviewer Expertise:**

5: The reviewer is absolutely certain that the evaluation is correct and very familiar with the relevant literature

**Robotics Focus:**

Sufficient demonstration on hardware

**Strengths And Weaknesses:**

## Strengths
- A framework for pre-training behaviors that can be composed to learn longer horizon tasks is a very promising approach and important to the community.
- The results and sim2real experiments look promising.
- The method itself, is a combination of existing work, but the overall approach is novel.

## Weaknesses
- The core contribution, specially with respect to existing work is not explicitly highlighted. There exists baselines, it is shown that they either fail to solve the tasks or perform poorly. However I had to briefly check those methods to decipher how they differ.
- Certain assumptions is made with regards to the availability of the sub-task trajectories, and how *exactly* might match to the final task scenario.

**Summary Of Recommendation:**

The core underlying assumptions using this method is that they consider  tasks that require the robot to compose previously seen behaviors (e.g., pushing, grasping, and opening) in order to perform a temporally extended task that sequences these
behaviors in a new way (by mix and matching these skills).

This method performs pre-training in order to learn a lossy representations during the offline pre-training process for the goal-conditioned policy using a variational information bottleneck (VIB). Such representation learns to capture the minimal information needed to determine how a particular goal can be reached.
The learned affordance models, captures the distribution of reachable future states and recursively propose subgoals conditioned on the initial state of each task. The affordance model is a generative model $m(s'|s.u)$ where $u$ is the latent representation that captures information of the transition from current state to the goal s'. Its trained as a conditional variational decoder (CVAE) using evidence lower bound (ELBO) objective function. By adding a a variational information bottleneck (VIB) to the reinforcement learning objective, which constrains the mutual information $I(s_t; z_t)$ and $I(g; z_g)$ between the state space and the representation space. This is converted to an an unconstrained optimization problem by applying the bottleneck on the Q-function representation and directly selecting the Lagrange multiplier $\alpha$
resulting to the VIB objective.

---

> ### Author Response · Authors · 2022-08-23
> **Author Response to Reviewer xMTo**
>
> Dear Reviewer,
>
> Thank you very much for your informative and constructive comments.
>
> 1. **Core contributions with respect to prior work.**
>
>     Our core contribution is a framework that leverages broad offline data for new temporarily extended tasks using learned lossy representations of states and goals. The three baselines compared in the paper also use subgoals to solve long-horizon tasks. LEAP learns a variational auto-encoder (VAE) to capture the prior distribution of states and plans for subgoals without conditioning on the initial state. GCP learns a goal predictor that hierarchically generates intermediate subgoals between the initial state and the goal state. PTP learns an affordance model to recursively generate a sequence of subgoals conditioned on the initial state. Our method also uses an affordance model during fine-tuning. But instead of generating and planning subgoals in the original state space, our method learns a representation of goals to facilitate generalization across domains. We will add such an explanation in the paper. A more detailed discussion of our method compared to baseline methods can be found in Sec. 5 (Line 311).
>
> 2. **How are the trajectories plotted in Figure 5?**
>
>     The 9 trajectories are selected from the offline dataset. As described in Sec. 4.3, we first run t-SNE on the whole offline dataset and project each image observation to a 2D point as shown in Figure 5. Then we connect the 2D points corresponding to the image observations in each of the 9 trajectories in the chronological order.
>
> 3. **Difference between Ours(Broad data) and Ours(Target Data).**
>
>     As we discussed in Sec. 4.2, when using only demonstrations in the target scene without broad offline data, both the pre-trained policy and affordance model do not generalize well. This is in accord with the findings in the original paper that introduces Bridge Data [1]. As a result, the planned subgoals are often not informative enough and the policy cannot reach the planned subgoals most of the time. Therefore, the policy often fails to collect sufficient useful data to improve the performance during online fine-tuning in this case.
>
> 4. **Questions about the state space.**
>
>     The state space does not include the proprioceptive state of the robot or the task. The current network architecture of the state encoder is designed for a fixed input dimension and cannot be used for environments of different image dimensions. We agree with you that this is however not a huge issue if we rescale the images to the same size or use alternative network architectures that are agnostic to input dimensions.
>
> 5. **Examples of how lossy representations abstract away task-irrelevant factors.**
>
>     As shown in Figure 5, the trajectories are clustered in the learned lossy representation space based on the semantics of the behaviors instead of task-irrelevant factors such as camera angles, objects and illumination conditions. As a comparison, in the VQVAE representation space, image observations are grouped together only when they have the same camera angles, objects, and illumination condition, while most semantically similar trajectories are far apart from each other.
>
> 6. **Assumptions about the availability of sub-task trajectories and how exactly they match to the target tasks.**
>
>     FLAP can be applied as long as the offline dataset and the target environment share the same state and action space. However, we agree that if the dynamics of the testing environment is out of the distribution of the offline data, it might be infeasible to acquire any generalizable policy, value functions, or affordance model through pre-training on the offline dataset. But the learned state encoder can still be potentially useful for the target task. How to extend FLAP for settings where the target tasks are significantly different from the offline data will be an exciting direction to explore in the future work.
>
> 7. **Overview of IQL.**
>
>     Thank you for the suggestion. We will add an overview of IQL in the supplementary.
>
> We hope these additional explanations have addressed your previous questions. Please don’t hesitate to let us know if you have any other questions or comments.
>
> [1] [Bridge Data: Boosting Generalization of Robotic Skills with Cross-Domain Datasets
> ](https://arxiv.org/abs/2109.13396)

---

### Official Review · Reviewer_jYk6 · 2022-07-27

**Originality:** Very Good
**Technical Quality:** Very Good
**Clarity Of Presentation:** Very Good
**Impact:** 4

**Recommendation:**

Strong Accept: I recommend accepting the paper and will argue for my recommendation even if other reviewers hold a different opinion.

**Summary:**

This work uses a big video dataset to train an affordance model, which can be used to rollout virtual trajectories on a learned feature space. In addition, they design a cost function to evaluate the virtual trajectories' quality. With the help of above-mentioned tools, they can plan a high-quality path from the current state to the desired target on the learned feature space, and use the future states on the path as the subgoals to guide a goal-conditioned policy to control the robot. Their experiments show that their method can be viewed as a good offline-pretrained model. After online finetuning in new tasks, their model outperforms baselines by a large margin.

**Issues:**

I’m curious about the required computational resources. As the model is trained with a big video dataset, and usually training with video is costly, it is important to describe the computational resource they use to train the model. For example, the number of GPUs, the training time, etc.

It will also be definitely interesting (but not necessary) to see if their method can be applied to the setting where the test robot arm and the robot arm in the offline dataset are different, as other labs or companies might not have the same robot arm as the dataset, and collecting new datasets are costly.



**Quality Of The Limitations Section:**

Limitations are addressed clearly

**Reviewer Expertise:**

3: The reviewer is fairly confident that the evaluation is correct

**Robotics Focus:**

Sufficient demonstration on hardware

**Strengths And Weaknesses:**

I think the research direction that utilizes a large offline dataset to boost robotic performance is very promising. This paper introduces a novel and interesting method in this direction. Their theory looks reasonable. Their experiments are done on real robots and look solid and significant. The paper is well written and easy to follow.

In the limitation discussion, they briefly describe the importance of the similarity between the offline data and the online tasks. I think it is better to have some experiments to show how the similarity affects the performance. For example, create some target tasks with different similarity levels w.r.t. offline data, and analyze the failure cases.


**Summary Of Recommendation:**

I think this paper proposes a novel and effective method in a very important research direction without obvious issues. Their experiment looks convincing. Therefore, I tend to accept this paper.

**After Authors' Replies**
The authors addressed my questions reasonably and I'm satisfied with the research topic they target at, the novelty of their method, and their experiments. Therefore, I think this paper should be accepted and I increased my score.

---

> ### Author Response · Authors · 2022-08-23
> **Author Response to Reviewer jYk6**
>
> Dear Reviewer,
>
> Thank you very much for your informative and constructive comments.
>
> 1. **Evaluation on target tasks of different similarity levels.**
>
>     As shown in Figure 3, the three real-world target tasks evaluated in the paper are actually listed in ascending levels of task difficulties and similarities to the offline data. The offline dataset contains demonstrations of single-stage interactions mostly on the same table surface or in the same sink. In Task A, the robot is asked to perform a single-stage pick-and-place tasks, which are very similar to the demonstrated trajectories in the offline dataset. In Task B, reaching the goal requires multi-stage interactions with objects on the same table surface. In Task C, the tasks are multi-stage and the objects are moved between the table surface and the sink. As shown in Table 1, we found that the success rates of Task A and B are higher than Task C. Qualitatively, we found that the most of the failures happen when the robot is trying to transit between stages or different workspaces, which are rarely seen in the pre-collected offline dataset. Videos of these target tasks can be found in the supplementary video and the project webpage included in the paper (https://sites.google.com/view/project-flap).
>
> 2. **Computational resources and training time.**
>
>     We use 1 TITAN RTX GPU with 24GB of memory and 1 Intel Xeon Skylake 6130 CPUS with 48GB of memory during training. The pre-training takes around 10 hours in total and the fine-tuning takes around 2 hours for each target task. We will include this information in the supplementary material as the reviewer suggested.
>
> 3. **Future application to settings where training and testing robots are different.**
>
>     We appreciate the reviewer’s suggestion and we agree that this would be an interesting future direction to explore. As long as the offline dataset and the target environment share the same state and action space, FLAP can be applied without any changes in the algorithm. However, if the dynamics of the testing environment is out of the distribution of the offline data, the pre-trained policy, value functions, and affordance model might not generalize. But the learned state encoder can potentially still provide useful features in the target environment. How to transfer the learned knowledge to completely different robotic systems would be an exciting research topic to study in the future work.
>
> We hope these additional explanations have addressed your previous questions. Please don’t hesitate to let us know if any further clarification would help you reconsider your rating.

---

> > ### Comment · Reviewer_jYk6 · 2022-08-25
> > **Reply**
> >
> > Thank you for your answers to my questions. It solves my concerns. I'll increase my score.

---

### Official Review · Reviewer_qYWX · 2022-07-29

**Originality:** Good
**Technical Quality:** Good
**Clarity Of Presentation:** Good
**Impact:** 3

**Recommendation:**

Weak Accept: I recommend accepting the paper, but will not argue for my recommendation if the majority of other reviewers have a different opinion.

**Summary:**

This work investigates how to make use of large offline data for fast online adaptation in a goal-conditioned RL setting.

The main idea consists of:
1) Learn a low dimensional representation for state/goal via an encoder (e.g. $z_s = \phi(s)$ and $z_g=\phi(g)$). To learn a robust representation, the authors propose to use information bottleneck, that minimize the mutual info between $z$ and $s$ while optimizing the RL objective.
2) Learn an "affordance" model, basically an inverse dynamics model that given a pair $(s, s')$, predicts what "action" leads to $s'$ given $s$. The model is formulated as a conditional variational autoencoder (CVAE), where the latent $u$ plays the role of "action".
3) Given an initial state $s$ and goal $g$, project them to latent space learned in 1), then using the learned affordance model in 2) to solve a planning problem. Additionally, some online data and reward shaping help fine-tune the performance.

**Issues:**

1. Please go through equations again and answer my question regarding the ELBO.
2. In Line 192, the author mentioned switching goals every $h$ steps. Goal switching is an important problem in option based learning works, but here it is assumed to using a heuristics. Why a constant $h$ can work here?
3. Maybe I miss this part, but can the author provide some descriptions about how they choose hyperparameters and how hard to tune them?

**Quality Of The Limitations Section:**

Limitations are addressed clearly

**Reviewer Expertise:**

3: The reviewer is fairly confident that the evaluation is correct

**Robotics Focus:**

Highly relevant to robotics but no hardware experiments

**Strengths And Weaknesses:**

***Strengths***:
1. The paper is well-written and the main idea is easy to follow.
2. Each proposed step is straight-forward and reasonable to my opinion.

***Weaknesses***:
1. Decomposing everything to 3 separate steps plus online fine-tuning is a bit cumbersome, especially for every new task, the model needs to fine-tune everything except the encoder.
2. While I have no doubt about step 1 and 2, I am a bit concerned about how practical is step 3. First, step 3 involves hyper-parameters like $V_\text{min}$, $V_\text{max}$, $p_\text{min}$, $\eta_{1,2,3}$, all of which have to be manually picked, possibly even for every new task. Also, I am a bit confused why in the constraint, we need both constraint on $V$ and constraint on $u$. It seems to me that both are playing a similar role.
3. Moreover, as already mentioned by the authors, the assumption of generalization is strict, the new environment has to be very similar to the old ones for meaningful generalization. Given the heavy procedure, I am not sure whether the "model-based" approach here is worthy.

***Other points***:
1. Line 34, you mention that the goal is to solve temporal extended task in zero-shot fashion. But isn't the proposed work in a few-shot setting?
2. Should CVAE be conditional variational **auto**encoder (line 89/167)? autoencoder is very different from encoder alone.
3. Equation 4 seems problematic to me (please correct me if I am wrong). Here, $z_g$ should be $z'$? Also, if you take expectation of $z' \sim \phi(s')$ outside, then it should not be a KL inside?

**Summary Of Recommendation:**

I think this paper proposes a reasonable pipeline for solving temporal extended tasks in a model-based fashion. The part I like is that it assumes GCRL setting, where manually designed reward is not necessary. But as I mentioned in the weakness section, I think there is some concern about how practical the method is. In general, I think the pros outweigh the cons so I recommend weak accept.

---

> ### Author Response · Authors · 2022-08-23
> **Author Response to Reviewer qYWX**
>
> Dear Reviewer,
>
> Thank you very much for your informative and constructive comments.
>
> 1. **Fine-tuning everything except for the encoder for every new task.**
>
>     We would like to clarify that both the encoder and the affordance model are fixed during online fine-tuning. As described in Line 57 and 121, only the policy along with the value function and the Q function are fine-tuned for each task. This is the same as in the prior work [1].
>
> 2. **Decomposing everything into 3 separate steps plus online fine-tuning.**
>
>     Instead of having 3 steps plus online fine-tuning, FLAP consists of an offline pre-training phase and an online fine-tuning phase as described in Sec. 3. During offline pre-training, the state encoder, the policy, the value functions, and the affordance model are trained. During online fine-tuning, the subgoals are planned in each new episode conditioned on the initial state and the final goal. The planning is part of the online fine-tuning phase, instead of a separate step conducted only once before the fine-tuning phase.
>
> 3. **Hyperparameter tuning.**
>
>     We would like to clarify that instead of tuning the hyperparameters differently for each new task, the exact same set of hyperparameters are used for all target tasks (the details are described in Sec. B of the supplementary). We use the same IQL hyperparameters as in [1]. We choose$\alpha$ and $\beta$ through grid search in simulation and the same hyperparameters are used in the real world. An ablation study can be found in Sec. C.
>
> 4. **Questions about the cost function of the planner.**
>
>     We use the same set of planning hyperparameters for all three target tasks. Both the constraint on the values and $u$ intend to encourage the planned subgoals to be feasible, but their specific purposes are different. While the constraint on $u$ prevents the affordance to propose subgoals that are out of the distribution of the offline data, the constraint on the values encourages the chosen subgoals to be reachable by the trained policy. In practice, we found including both constraints leads to the best performance.
>
> 5. **The sentence in Line 34.**
>
>     This sentence intends to motivate the necessity of online fine-tuning by explaining that the pre-trained policy cannot solve temporally extended tasks in zero shot in the target environment. Our method is not in a zero-shot setting, but we agree with you that the online fine-tuning phase for each target task in FLAP can be considered as few-shot. We will add the following sentence after Line 34 to avoid further confusion: *“To solve the target tasks, we would need to transfer the learned knowledge to the testing environment and efficiently fine-tune the policy in an online manner.”*
>
> 6. **Time budget for switching subgoals.**
>
>     We switch to the next subgoal in two conditions: (1) when the current subgoal is reached, i.e. $d(s - s_g) \leq \epsilon$, the subgoal, and (2) if the current subgoal is not reached within a time budget of $h$. To choose the suitable $h$, the major concern here is to make sure $h \gt \Delta t$ so that the goal-conditioned policy will have enough time to reach the next subgoal, where $\Delta t$ (defined in Sec. 3.2) is the time interval for training the affordance model. In our implementation, we set $h = 2 \Delta t$ to spare a more generous time budget for the trained policy. We will make this detail clearer in the paper.
>
> 7. **Typo in the ELBO.**
>
>     Thank you for pointing out the typo. Eqn.4 is supposed to be
>     $\mathbb{E}_{z \sim \phi, u \sim q}[ KL( m(z' | z, u) || \phi(z' | s') ) ]$
>
>     $- \beta \mathbb{E}_{z, z' \sim \phi} [ KL ( q(u| z, z') || p(u) ) ]$, where $KL(\cdot || \cdot)$ denotes the KL-divergence.
>
> 8. **Typo for the CVAE.**
>
>     CVAE in both places should be *Conditional Variational Autoencoder*.
>
> 9. **Highly relevant to robotics but no hardware experiments.**
>
>     We believe that sufficient hardware experiments have been demonstrated in the paper. To conduct thorough comparisons, we evaluate our full method, the baseline methods, and an ablation on three different target tasks. The fine-tuning and evaluation for these comparisons totally took around 40 hours in the toy kitchen environment with a real-world WidowX Robot Arm. The videos of both our real-world and simulated experiments can be found in the supplementary video and the project webpage included in the paper (https://sites.google.com/view/project-flap).
>
> We hope these additional explanations and modifications have addressed your previous questions. Please don’t hesitate to let us know if any further clarification would help you reconsider your rating.
>
> [1] [Planning to Practice: Efficient Online Fine-Tuning by Composing Goals in Latent Space.](https://arxiv.org/abs/2205.08129)

---

> > ### Comment · Reviewer_qYWX · 2022-08-26
> > **Response to Authors**
> >
> > I appreciate the authors' clarifications, which resolved my concerns. My summary of recommendation remains the same, because of the reasons I mentioned in my original review.

---

### Meta-Review · Area_Chair_Wpoj · 2022-08-15

**Recommendation:** Accept (Oral)
**Confidence:** 4

**Metareview:**

Below is a summary of the strengths and weaknesses of the paper, according to the reviewers.

Strengths:
- The proposed method is novel and interesting.
- The proposed method performs well in experiments.
- Real-world experiments are conducted.
- The paper is well written and easy to follow.

Weaknesses:
- In Step 3, there are a number of hyper-parameters. If these hyper-parameters must be tuned for every new task, then the method could be very impractical.
- The offline data and data for the target task must come from very similar tasks. This is already highlighted by the authors in the paper's limitations section, so it is not a major weakness in the paper itself. However, it would have been helpful to see experiments exploring how different levels of similarity affect performance.
- It is not always clear what the core difference is between the method and the baselines in the evaluation.

In your rebuttal, please address the above weaknesses, as well as any other concerns or questions raised by the reviewers.

---------

After rebuttal:

Reviewers retained their strong opinion of the paper following the rebuttal. This is topical paper focussing on offline reinforcement learning, with a novel technical contribution and interesting real-world experiments for evaluation. One key limitation -- that the offline data needs to be from a similar distribution to data in the target domain -- is a general current limitation of the field of offline reinforcement learning, rather than a specific limitation of this paper, and is an open problem. Therefore, there are no major weaknesses in this paper, and it clearly provides an important contribution to the field.

**Best Paper Nomination:**

No

---

> ### Author Response · Authors · 2022-08-23
> **Author Response to Area Chair Wpoj**
>
> Dear Area Chair,
>
> Thank you so much for the constructive suggestions and comments! We appreciate the recognition of our technical novelty, clarity, and experimental results. We address the listed issues below:
>
> 1. **Hyperparameter tuning.**
>
>     We would like to clarify that instead of tuning the hyperparameters differently for each new task, the exact same set of hyperparameters is used for all target tasks. We tune the hyperparameters once through grid search in simulation and use them for both the simulation and the real world. A detailed description of hyperparameters can be found in Sec. B of the supplementary. As shown in the supplementary and the project webpage (https://sites.google.com/view/project-flap), the same hyperparameters enable FLAP to solve various target tasks in simulation and the real world, which we believe have demonstrated the practicability of our method.
>
> 2. **Evaluation of target tasks of different similarity levels.**
>
>     The three real-world target tasks evaluated in the paper are actually listed in ascending levels of difficulties and similarities as shown in Figure 3. The offline dataset contains demonstrations of single-stage interactions with objects mostly on the same table surface or in the same sink. In Task A, the robot is asked to perform single-stage pick-and-place tasks, which are very similar to the demonstrated trajectories in the offline dataset. In Task B, reaching the goal requires multi-stage interactions with objects on the same table surface. In Task C, the tasks are multi-stage and the objects are moved between the table surface and the sink. As shown in Table 1, we found that the success rates of Task A and B are higher than Task C. Qualitatively, we found that most of the failures happen when the robot is trying to transit between stages or different workspaces, which are rarely seen in the pre-collected offline dataset.
>
> 3. **The core difference between our method and the baselines.**
>
>     Our core contribution is a framework that leverages broad offline data for new temporarily extended tasks using learned lossy representations of states and goals. The three baselines compared in the paper also use subgoals to solve long-horizon tasks. A detailed explanation of the difference between our method and baseline methods can be found in Sec. 5. LEAP learns a variational auto-encoder (VAE) to capture the prior distribution of states and plans for subgoals without conditioning on the initial state. GCP learns a goal predictor that hierarchically generates intermediate subgoals between the initial state and the goal state. PTP learns an affordance model to recursively generate a sequence of subgoals conditioned on the initial state. Our method also uses an affordance model during fine-tuning. But instead of generating and planning subgoals in the original state space, our method learns a lossy representation of goals to facilitate generalization across domains. We will make this clearer in the paper.
>
> We hope these additional explanations have addressed the questions above. Please don’t hesitate to let us know for any additional comments or questions.